# Psychometric Evaluation of the Chinese Version of Mild Cognitive Impairment Questionnaire among Older Adults with Mild Cognitive Impairment

**DOI:** 10.3390/ijerph20010498

**Published:** 2022-12-28

**Authors:** Qingmin Dai, Hong Su, Zanhua Zhou, Caifu Li, Jihua Zou, Ying Zhou, Rhayun Song, Yang Liu, Lijuan Xu, Yuqiu Zhou

**Affiliations:** 1Ecology College, Lishui University, No. 1 Xueyuan Road, Lishui 323000, China; 2Department of Nursing, Daqing Campus, University of Harbin Medical, 39 Shinyo Road, Daqing 163319, China; 3Medicine College, Lishui University, No. 1 Xueyuan Road, Lishui 323000, China; 4Nursing College, Chungnam National University, 266 Munwha-dong, Daejeon 35015, Republic of Korea

**Keywords:** mild cognitive questionnaire, older adults, psychometric test, quality of life, reliability, validity

## Abstract

Background: There is a lack of instruments for measuring quality of life (QOL) in Chinese patients with mild cognitive impairment (MCI). This study aimed to translate the Mild Cognitive Questionnaire (MCQ) into the Chinese language and to evaluate the reliability and construct validity of the MCQ-Chinese among older adults with MCI. Methods: Linguistic translation and validation of the questionnaire were conducted according to the MCQ developer and Oxford University Innovation guidelines. After a pilot test, the final version of the MCQ-Chinese was applied to a convenience sample of older adults with MCI (n = 186). Cronbach’s alpha and confirmatory factor analyses were used to assess the reliability and construct validity of the MCQ-Chinese. In addition, non-parametric analysis was used to assess convergent and discriminant validity. Results: The total scale and all the factors had good internal consistency, with Cronbach’s alpha values ranging from 0.90 to 0.92. Confirmatory factor analysis indicated satisfactory goodness of fit for the 2-factor MCQ. The MCQ-Chinese had a good convergent validity, and the discriminant validity was confirmed with a significant difference in MCQ scores in different health conditions. Conclusions: MCQ-Chinese is a reliable tool for assessing QOL among Chinese older adults with MCI.

## 1. Introduction

With a rapidly aging population, there has been an increase in the number of elderly persons diagnosed with mild cognitive impairment (MCI) [1]. Because of cognitive pathology, older adults with MCI have a lower quality of life (QOL) than those who are cognitively healthy [2,3]. In previous studies, QOL was mostly assessed using a generic instrument. Recently, it has been shown that the Mild Cognitive Impairment Questionnaire (MCQ) is a valid and reliable instrument for evaluating QOL among older people with MCI [4,5,6]. There is a need to adapt the questionnaire transculturally and validate it for use in Chinese.

In China, by the end of 2020, the population of older adults aged above 60 years reached 0.26 billion, accounting for 18.7% of the total population [7]. A recent national cross-sectional study [1] indicated that the overall adjusted MCI prevalence was 15.5% (95% CI 15.2–15.9) among older Chinese people aged above 60 years, and the prevalence increased with age. MCI is a clinical pathology that frequently precedes the development of dementia, referring to impairment in cognition that is below normal age-related cognitive decline but not severe enough to cause significantly impaired daily function [8]. MCI is considered to be a transitional stage between normal aging and dementia that causes irreversible decline in physical, cognitive, and social function [9]. The annual conversion rate from MCI to dementia is approximately 15% in community samples and 34% in clinical samples [10], and the probability of developing dementia within five years is as high as 70% [11].

MCI is a stage that can determine the onset of clinical cognitive pathology; therefore, early control and management of MCI are very important [12]. Recently, an increasing number of studies have explored the QOL and experiences of people with MCI. A qualitative study indicated that people with MCI experienced cognitive changes (loss of initiative, difficulty in concentrating), impairment in daily activities, uncertainty about their abilities and environmental reactions, and the need to adopt coping strategies with memory aids or repetition [13]. Subjective memory complaint (SMC) is a basic feature of MCI; thus, it is a very common phenomenon in the daily lives of older adults with MCI [14]. Previous studies have indicated that SMC is associated with depressive symptoms and problems in activities of daily living [15,16]. Therefore, the subjective assessment of cognitive impairment and emotional and functional problems is an important aspect of the QOL for patients with MCI. In previous studies, generic instruments, such as SF-36 [3,17], World Health Organization Quality of Life Scale-Brief Form (WHOQOL-BREF) [2,18], EuroQol five dimensions questionnaire (EQ-5D) [19], and World Health Organization Quality of Life Instrument-Older Adults Module (WHOQOL-OLD) [2,18] were used to assess the QOL of patients with MCI. These instruments are all multidimensional and measure QOL domains such as physical, social, and mental functions, and aspects such as vitality, environment (e.g., home environment, healthcare services, financial resources, and transport), pain, self-care, and usual activities [20,21,22]. The SF-36 is one of the most-common health-related QOL instruments, but it has been described as being difficult for the older people to complete [23]. The EQ-5D focuses on whether an individual can perform their usual activities and self-care [21]. Both the WHOQOL-BREF and WHOQOL-OLD originate from the WHOQOL-100 assessment. WHOQOL-BREF includes the four dimensions of physical, psychological, social, and environmental health [22] while the WHOQOL-OLD includes the domains ‘sensory abilities’, ‘autonomy,’ ‘past, present, and future activities’, ‘social participation’, ‘death and dying’, and ‘intimacy’ [20]. However, the above-mentioned instruments are not able to assess cognitive problems.

Disease-specific instruments produce more targeted results than generic instruments. A review identified 15 dementia-specific QOL instruments, which included items pertaining to mood, self-esteem, social interaction, and enjoyment of activities [24]. As people with MCI have less functional and cognitive impairment than those with dementia, the issues of QOL in the former are different from the latter. For example, the QOL-AD includes five domains: perceived QoL, behavioral competence, psychological status, interpersonal environment, and physical functioning, and most of them are objective indicators [25]. Therefore, the measures developed to assess the QOL in patients with dementia are not valid for use in MCI. In addition, the instruments to measure QOL of those suffering from dementia are too long, such as health-related QOL for people with dementia with a 28-item self-rated and 31-item proxy-rated version; it was recommended that the two versions be used together [26].

For people with MCI, a simple and short self-rated questionnaire was suggested [27]. This 13-item MCQ instrument used to assess the QOL in people with MCI, contains two domains of emotional and practical concerns related to cognitive impairment [5]. It is a simple instrument that can be easily used in various clinical and research settings. Previous studies have shown good reliability and construct validity in the English, Spanish, and Korean versions [4,6,28]. However, it has not been translated or adapted to the Chinese language or context. The purpose of this study was to translate the MCQ into Chinese, to evaluate its reliability and construct validity on the local population and culture of China, and to provide an effective tool to assess the QOL of older adults with MCI.

## 2. Materials and Methods

### 2.1. Design

A methodological and cross-sectional correlation study was conducted.

### 2.2. Participants and Settings

Using convenience sampling methods, we recruited 186 older people in five aged centers in Lishui, Zhejiang Province, China. The inclusion criteria were that participants be 75 years or older, without a communication disorder or hearing impairment, and diagnosed with MCI with the use of the Chinese version of the Montreal Cognitive Assessment (MoCA). The response rate of the 210 distributed questionnaires was 95% (n = 200), and after screening patients with MCI using the MoCA, 186 remained for data analysis. The sample sizes were estimated following the recommendation of Polit and Beck [29]. For testing a scale, the sample size should be 10 times the number of items in the tested scale. With 13 items in the MCQ and a response rate of 80%, there was a need for 163 participants.

### 2.3. Instruments

#### 2.3.1. MCQ

The Mild Cognitive Impairment Questionnaire (MCQ) [5], which assesses the quality of life of people with mild cognitive impairment, contains 13 items that assess two factors: practical concerns (7 items, Cronbach’s alpha (α) = 0.85) and emotional concerns (6 items, α = 0.91). Response alternatives for each item are rated on a 5-point scale, with 0 for none and 5 for always. For scores of the total scale and factors, items are summarized, divided by the highest possible factor score, and multiplied by 100. The highest possible score is 100, with higher scores indicating worse QOL. We obtained permission from the developer to translate the MCQ into Chinese.

To translate the original English version of the (MCQ) into Chinese, we followed the steps in the guidelines of translation and linguistic validation process as per the clinical outcome assessment [30]. First, two native bilingual speakers independently translated the original MCQ into Chinese. Both translators had geriatric nursing backgrounds and were provided with the concept elaboration document for the MCQ. One of the two translators had a PhD, and the other was a PhD candidate. The two forward translations were sent to the first author of the manuscript, the project manager, and an agreement on the final version was reached after reconciliation with the two forward translators. Second, two other translators who were Chinese-Singaporean PhD candidates without a medical background were asked to back-translate the reconciled language version of the MCQ into English independently. The two back-translations were sent to the owner of the MCQ from Oxford University Innovation and were reviewed and compared with the original version to highlight any discrepancies in meaning or terminology. A comment on item 5 was obtained from Oxford University Innovation. Third, it was clarified that the meaning of the concept of “slowed down” was as “you cannot think as quickly as you would like,” not “you cannot behave as quickly as you would like to”. For revised item 5, forward and backward translations were conducted again. Finally, all authors reviewed the translations and reached a consensus, and a pre-final version was produced. According to the guidelines of clinical outcome assessment [30], the pre-final version of the Chinese MCQ was pilot-tested among five older adults aged >75 years with mild cognitive impairment screened using MoCA. The MoCA scores ranged from 15 to 24 for the five participants of the pilot test. The test indicated that the older adults spent 3 to 5 min to answer the questionnaire. In the test, participants were asked whether the questions in the Chinese version of the MoCA were easy to understand and answer, and it was revealed that participants understood the questions easily.

#### 2.3.2. Chinese Version of WHOQOL-OLD

The Chinese version of the WHOQOL-OLD [31] consists of 24 items with 6 dimensions: sensory abilities (4 items, α = 0.84); autonomy (4 items, α = 0.71); past, present, and future activities (4 items, α = 0.76); social participation (4 items, α = 0.71); death and dying (4 items, α = 0.84), and intimacy (4 items, α = 0.82). Response alternatives for each item are rated on a 7-point scale, with 1 = not at all and 5 = an extreme amount. Higher scores indicate better QOL. To be more readily incorporated into routine care, three short forms with the six best items of the original WHOQOL-OLD [32] were developed. The three short forms were tested among older adults from 20 international centers including China, and they were found to have good reliability and validity.

#### 2.3.3. FRAIL Assessment

FRAIL scale [33] was used to assess frail status. FRAIL includes five items: fatigue, resistance, ambulation, illnesses, and weight loss. Participants without symptoms/signs were identified as robust, with one or two were identified as prefrail, and those with at least three were identified as frail.

#### 2.3.4. Beijing Version of Montreal Cognitive Assessment

The Beijing version of the Montreal Cognitive Assessment (MoCA) [34] was used to screen older adults with mild cognitive impairment. The MoCA has high sensitivity and specificity for detecting MCI [35]. The MoCA consists of seven domains: visuospatial/executive function (five scores), naming (three scores), attention (six scores), language (three scores), abstraction (two scores), delayed memory (five scores), and orientation (six scores). The total score ranges from 0 to 30, with higher scores indicating better cognitive function. Participants with a total score lower than 26 were diagnosed as having mild cognitive impairment. We obtained permission to use the Beijing University version of MoCA.

### 2.4. Data Collection and Ethical Consideration

Data were collected from November 2021 to January 2022. The data collectors were provided sufficient training by the project manager, the first author. For MoCA screening for MCI, all data collectors were trained using the MoCA instructions. The medical ethics committee of Lishui University, to which the first author is affiliated, approved this study (no. 20210031). For each aged center, the staff assembled older adults in one room to receive oral information regarding the study. Before data collection, all participants received oral and written information about the study content and signed an informed consent form. The form made it clear that participation in the study was voluntary, the participants had the right to withdraw from the study at any time, and it would not affect their social support and healthcare. The participants were also assured that their responses would be kept private and confidential, and the researchers and participants were not related. Toothpaste and soap were offered to the participants as gifts for participating in the study.

### 2.5. Data Analyses

We analyzed the data IBM SPSS Statistics 26.0 and IBM SPSS Amos (IBM Corp., Armonk, NY, USA). Internal missing data in MCQ, WHOQOL-OLD, MoCA, and FRAIL (a total of 13 missing data) were replaced with the participant’s median value in that factor.

Descriptive data are reported as mean, standard deviation (SD), median, frequency, and percentage. Internal consistency for the factors and the total scale of the MCQ were measured using Cronbach’s alpha coefficient. Confirmatory factor analysis (CFA) was conducted to evaluate construct validity. Some error terms within the same factor were allowed according to the results of the modification indices. The main indices in CFA included absolute parameters, such as chi-square/degree of freedom (x^2^/df) (<3 indicating acceptable), goodness of fit index (GFI) (>0.90 indicating acceptable) [36], and root-mean-square error of approximation (RMSEA) with 90% confidence intervals, with <0.08 indicating acceptable [37]. In addition, relative parameters, such as the comparative fit index (CFI) (>0.95, indicating good; >0.90, acceptable) [36], were included. According to the Shapiro–Wilk normality test (*p* < 0.05), visual inspection of normal q-q plots indicated MCQ with serious deviations from a normal distribution. To present the MCQ measuring QOL, convergent validity testing was conducted using Spearman correlation analysis between MCQ, the WHOQOL-OLD, and the three short forms of the WHOQOL-OLD. The correlation coefficient, with values more than 0.50, indicated a strong correlation [38]. Previous research has shown that health conditions, such as comorbidities and frailty status, are associated with QOL among older adults [3,39]. In the present study, discriminant validity was assessed using the Mann-Whitney U test, Kruskal-Wallis test (due to the distribution), and Bonferroni correction method to examine the difference in factor and total scores of MCQ among comorbidity and frail status groups. The criterion for statistical significance was set at *p* < 0.05.

## 3. Results

### 3.1. Descriptive Characteristics of the Participants

The average age of the participants was 79.5 (SD = 6.34, range 75–99), slightly more than half were men (52.7%), most had received primary school or lower education (74.7%), and most lived with family members (78%). Regarding health conditions, 42.5% of the participants had comorbidities, 41.4% were pre-frail, and 21 were frail. The total score for cognitive function was 15.92 (SD = 6.36), and the total score for QOL was 67.18 (SD = 14.84). Table 1 presents the participant characteristics and health conditions.

The score for total scale of MCQ was 41.99 (SD = 16.13), with a higher score for practical concerns (mean = 44.75, SD = 16.77) than emotional concerns (mean = 38.76, SD = 17.14) (Table 2).

### 3.2. Reliability

The Cronbach’s alpha coefficient was 0.96 for the total score of MCQ, and for the subscale ranged from 0.92 to 0.96 (Table 2).

### 3.3. Construct Validity

A two-factor model for the Chinese version of the MCQ was tested using CFA, according to the original version of the scale construct. With two factors correlated and some error terms within factors based on modification indices, the model fits were acceptable with a x^2^/df of 1.79, CFI and GFI higher than 0.90, and RMSEA = 0.063 (90% CI, 0.045–0.085). The standardized factor loadings for the items ranged from 0.69 to 0.95. The factors and items are presented in Table 2 and Figure 1. 

### 3.4. Convergent Validity

The Spearman correlation coefficients between the total score of the MCQ and WHOQOL-OLD and the three short versions of the WHOQOL-OLD ranged from −0.54 to −0.57, indicating a good convergent validity with a strong correlation. Significant correlation coefficients were also found between the MCQ and WHOQOL-OLD and the three short versions of the WHOQOL-OLD (see Table 3).

### 3.5. Discriminant Validity

Due to the distribution of MCQ, a non-parametric test was conducted to examine the difference in total subscale scores of the Chinese version of the MCQ among groups with and without comorbidity, and groups with different frailty status. For the total and subscale scores, the participants without comorbidity had a lower score (lower score indicating better QOL) than those with comorbidity, and the pre-frail or frail groups had a higher score than the robust group (Table 4).

## 4. Discussion

In this study, we translated and validated the Chinese version of the MCQ. The results indicated that the Chinese version of the MCQ has good internal consistency and satisfactory construct validity. The non-parametric analysis indicated that the Chinese version of the MCQ had good convergent and discriminant validity in distinguishing between healthy and unhealthy groups.

In the present study, the total score of MCQ was 41.99 (lower score indicating better QOL, SD = 16.13), the practical concern was 44.75 (SD = 16.77), and the emotional concern was 38.76 (SD = 17.14), the scores being higher than those in previous studies [4,5,6]. In the present study, the MoCA cognitive function score was 15.92 (with a lower score indicating higher cognitive function (SD = 6.36) while in the study by Song et al. [6], the MoCA cognitive function score was 17.78, and lower cognitive function was associated with a lower QOL. In addition, in the present study, the score of emotional concern was lower than practical concern (i.e., the emotional level was better than the practical level), and the results were different from those of previous studies. In the original study [5], and the Korean version [6], the score of practical concern was similar to that of emotional concern. In the present study, more than half of the participants were male, whereas in previous studies [4,5,6], almost 70% were female. A possible reason for this was that, compared to older men, older women reported higher levels of emotional problems [40,41]. Additional modifications were required during the translation process. In the present study, we followed the guidelines of translation and linguistic validation process for the Clinical Outcome Assessment, and only one comment about item 5 “slowed down” was received from the review of the original author. In the cognitive debriefing of the pilot test, it was shown that participants with MCI understood the items and responses in the MCQ-Chinese easily.

The Chinese version of the MCQ showed good internal consistency of the two dimensions, with Cronbach’s alpha coefficients, and all the total scores were above 0.90. The present study presented results similar to those of the original and Korean studies [5,6] The results of confirmatory factor analysis for the two-factor model with RMSEA of 0.065, CFI of 0.98, and GFI of 0.92 indicated satisfactory construct validity, which was close to the Korean and Spanish versions [4,6]. Correlation analysis revealed a strong negative correlation between WHOQOL-OLD and the three short forms of the WHOQOL-OLD. The findings demonstrated that the MCQ measured a trait similar to the WHOQOL-OLD, suggesting good convergent validity. Since our participants were aged above 75 years, WHOQOL-OLD, specifically designed to address the issues of older adults aged above 60 years, is a reliable and valid instrument to measure QOL [31,32]; WHOQOL-OLD and the short forms are valid measures to assess convergent validity. In addition, the similar correlation between the MCQ-Chinese version and short forms of the WHOQOL-OLD added evidence that the three short forms were equally effective in measuring QOL among older adults [32]. The non-parametric analysis showed that the participants with multimorbidity had significantly higher MCQ scores (indicating lower QOL) than those without multimorbidity, suggesting a good discriminant validity to distinguish between multimorbidity and no multimorbidity. In the present study, the pre-frail and frail participants reported higher scores on the MCQ (lower QOL) than the robust group, which was consistent with previous studies [42,43]. According to Arai et al. [44], cognitive frailty, which is an emerging concept, is defined as a condition of coexisting physical frailty and (MCI); therefore, the MCQ measure could be useful in understanding cognitive frailty.

### 4.1. Clinical Implication

These results have implications for clinical nursing. The current study obtained promising results regarding the psychometric properties of the MCQ when used among institutionalized older people. The fact that the MCQ-Chinese quickly provides a total score for QOL and the simplicity of its administration will facilitate its use to assess the effect of interventions for older adults with MCI. In addition, the significant difference in MCQ scores between the different frailty groups suggests that the MCQ-Chinese could be used on patients with cognitive frailty.

### 4.2. Methodological Considerations

This study has some limitations, such as the convenience sampling method that was used, and that data was collected from aged centers in one city in South China, which could influence generalizability. Another limitation was the inclusion criteria “being 75 years or older,” due to which some patients aged younger than 75 years were not included in the study. As it was the staff who assembled the older adults for the survey, there may have been gatekeeper problems in the data collection. For frailty, the missing data was rather substantial while they were only used to test the discriminant validity of the MCQ. The last limitation was that test-retest reliability was not determined for the MCQ-Chinese. This study has some strengths as well. In the present study, the response rate was high, and there were a few missing internal data points that were substituted with the median of the individuals for that factor. The CFA indicated satisfactory construct validity as well as good convergent and discriminant validity. The MCQ-Chinese had a high Cronbach’s alpha values for internal consistency, ranging from 0.92 to 0.96 for the total and subscales, respectively.

## 5. Conclusions

Our results indicated that the Chinese version of the MCQ is a useful, valid, and reliable tool for assessing the QOL of Chinese patients with MCI. As the MCQ is a simple, short, patient-rated instrument, its Chinese version could be used to measure the QOL of older adults with MCI. Further studies are needed to test the MCQ-Chinese in different clinical settings to confirm its stability.

## Figures and Tables

**Figure 1 ijerph-20-00498-f001:**
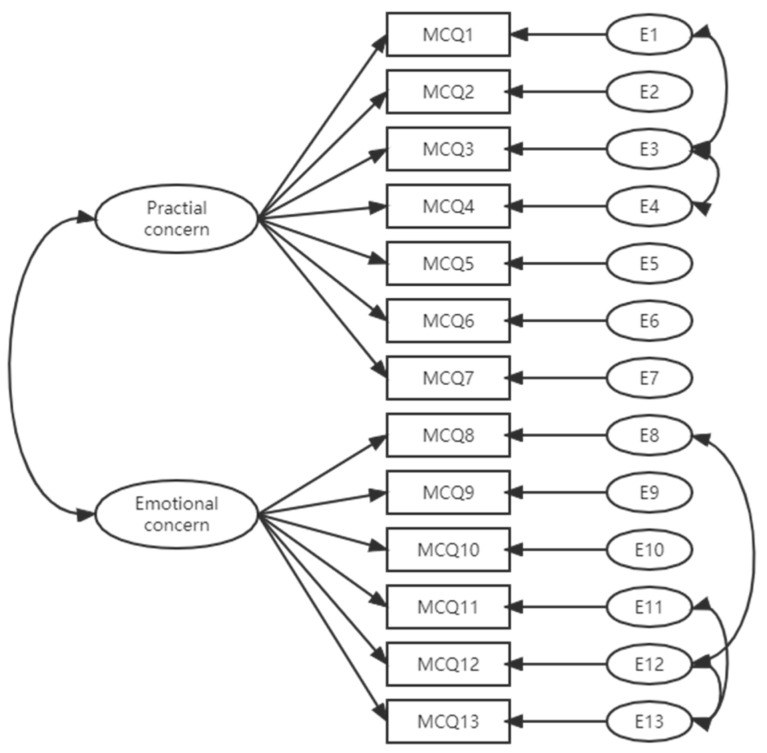
Flow chart presenting the underlying construct of Chinese version MCQ ([Mild cognitive impairment questionnaire] MCQ, [error term] E).

**Table 1 ijerph-20-00498-t001:** Sociodemographic characteristics and health conditions of participants (N = 186).

Variables	N (%)	Mean (SD) /Median (Q1:Q3))
Age		79.49 (6.34)
Number of children		2.50 (2.00:3.00)
Cognitive function (MoCA—Beijing Version)		15.92 (6.36)
Quality of life (WHOQOL-OLD)		67.18 (14.84)
Sex		
Male	98 (52.7)	
Female	88 (47.3)	
Marital status		
Married	124 (66.7)	
Divorced/single/widow (er)	62 (33.3)	
Living place		
In town	89 (47.8)	
In rural	97 (52.2)	
Living situation		
Living with family member	145 (78.0)	
Living alone	41 (22.0)	
Educational level		
Primary school or lower	139 (74.7)	
Middle school	28 (15.1)	
High school	16 (8.6)	
College or higher	3 (1.6)	
Source of income		
Retirement pension	107 (57.5)	
Children or others	79 (42.5)	
Level of income		
Low	89 (47.9)	
Middle	86 (46.2)	
High	11 (5.9)	
Main medical insurance type		
Employment medical insurance or self payment	28 (15.1)	
Basic residents medical insurance	158 (84.9)	
Comorbidity		
Yes	79 (42.5)	
No	107 (57.5)	
Frail status		
Robust	44 (23.7)	
Prefrail	77 (41.4)	
Frail	39 (21.0)	

**Table 2 ijerph-20-00498-t002:** The fit indices of Chinese version MCQ, scores for the items and factors and factor loadings in the confirmatory factor analysis (N = 186).

Chi-Square (df) *p*-Value	Chi-Square/df	CFI	GFI		SRMR	RMSEA (90%CI)
<0.001	1.789	0.98	0.92		0.036	0.065 (0.045–0.085)
**Factors/items (Cronbach’s Alpha)**			**Factors loading ML**	**Factors loading ML/** **Standardized**	**Mean (SD)**	**Median (Q1:Q3)**
Practical concerns (0.923)					44.75 (16.77)	42.86 (31.43:54.29)
1. Worry about forgotten things			1	0.69	2.62 (1.03)	3 (2:3)
2. Worry about sentence construction			1.08	0.74	1.99 (1.03)	2 (1:3)
3. Worry about forgetting plans			1.06	0.74	2.42 (1.03)	2 (2:3)
4. Worry about forgetting appointments			1.15	0.78	2.28 (1.04)	2 (1:3)
5. Worry about feeling slowed down			1.17	0.82	2.26 (1.01)	2 (1:3)
6. Worry about upsetting others because of memory problems			1.16	0.87	1.97 (0.94)	2 (1:3)
7. Feeling less independent			1.17	0.83	2.12 (1.01)	2 (1:3)
Emotional concerns (0.962)					38.76 (17.14)	40.00 (20.00:50.00)
8. Irritation or frustration about memory problems			1	0.95	1.88 (0.91)	2 (1:3)
9. Worry about memory			0.98	0.89	2.03 (0.96)	2 (1:2)
10. Feeling downhearted about memory			0.97	0.96	1.85 (0.88)	2 (1:2)
11. Worry about others’ reactions to memory problems			0.97	0.93	1.91 (0.90)	2 (1:2)
12. Worry about memory is worse than peers			0.94	0.85	1.92 (0.96)	2 (1:3)
13. Worry about memory worsening in the future			0.92	0.81	2.03 (0.99)	2 (1:3)

Abbreviations: Maximum likelihood (ML), Standard deviation (SD), Quartile (Q).

**Table 3 ijerph-20-00498-t003:** Spearman correlation between MCQ and WHOQOL-OLD.

Variables	Practical Concern	Emotional Concern	MCQ	WHOQOL-OLD Version 1	WHOQOL-OLD Version 2	WHOQOL-OLD Version 3	WHOQOL-OLD
Practical concern	--						
Emotional concern	0.80 **	--					
MCQ	0.96 **	0.94 **	--				
WHOQOL-OLD Version 1	−0.55 **	−0.48 **	−0.54 **	--			
WHOQOL-OLD Version12	−0.55 **	−0.51 **	−0.56 **	0.92 **	--		
WHOQOL-OLD Version 3	−0.56 **	−0.49 **	−0.55 **	0.88 **	0.90 **	--	
WHOQOL-OLD	−0.58 **	−0.51 **	−0.57 **	0.9 3 **	0.94 **	0.94 **	--

Note: ** Correlation is significant at the 0.01 level (2-tailed).

**Table 4 ijerph-20-00498-t004:** Discriminant validity of Chinese version MCQ.

Variables	Practical Concern Mean (SD)	Emotional Concern Mean (SD)	MCQ Total Mean (SD)
Comorbidity			
Yes	52.08 (17.14)	45.36 (17.79)	48.98 (16.66)
No	39.33 (14.32)	33.89 (14.94)	36.82 (13.65)
Mann-Whitney U	24.66	20.12	25.44
*p*-value	<0.001	<0.001	<0.001
Frail status			
Robust ①	34.41 (13.40)	29.09 (12.25)	31.96 (11.81)
Prefrail ②	47.61 (15.45)	41.47 (16.20)	44.78 (15.07)
Frail ③	55.16 (18.04)	49.23 (17.84)	52.43 (17.08)
Kruskal-Wallis	32.08	30.20	34.30
*p*-value	<0.001	<0.001	<0.001
Pairwise comparisons (*p*-value adjusted by Bonferroni correction)	② > ①; ③ > ①	② > ①; ③ > ①	② > ①; ③ > ①

Abbreviations: Standard deviation (SD), Mild cognitive impairment questionnaire (MCQ).

## Data Availability

The data that support the findings of this study are available on request from the corresponding author. The data are not publicly available due to privacy or ethical restrictions. Therefore, they are available from the corresponding author upon reasonable request.

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
