# Peer review of "Psychometric Evaluation of the Chinese Version of Mild Cognitive Impairment Questionnaire among Older Adults with Mild Cognitive Impairment"

_ijerph, 2022, doi:10.3390/ijerph20010498_

Round 1

Reviewer 1 Report

Introduction: I suggest to write the complete name of the WHOQOL-OLD the first time that it's mentioned, to facilitate the understanding of the acronym by the reader.

 Line 192: The definition of "frailty status" should be briefly explained here.

Metodology: It's important to include a paragraph describing how the researchers treated missing and atypical data.

Reviewer 2 Report

The authors present a validation study for the assessment of cognitive impairment, fulfilling the necessary assumptions to complete the work. However, I leave below some questions to be clarified.

1. title: appropriate 

2. Abstract: Contains necessary information for this type of study

3. Introduction: Consistent, objective and rational

4. Methods: Please improve your method by putting the following elements into sections, separate subsections: Study design; Participants and settings; Sample selection (inclusion and exclusion criteria; Sample size: (if you did not calculate the sample, explain why a sample of 186 individuals selected by convenience is considered sufficient for validating the instrument, taking into account the size of the population in China).

5. Results: I think it would be interesting to show the characterization data of the sample according to gender; I believe that the correlation is an assumption of the AFC, therefore, these data must be presented before the figure of the AFC. However, if I'm wrong, please correct me as my reference sources don't make this clear.

6. Discussion: Organized, excellent level of presentation of sections and subsections.

Reviewer 3 Report

Dear authors,

congratulations on the article and its structure. In order to improve it further, I recommend that you strengthen the introduction a bit more and add some analysis in relation to discriminant validity. 

Otherwise, I believe the manuscript is a suitable paper for this special issue and of high impact for the scientific community. Please also add as much information as possible regarding the ethical collection of data. 
